# The Role of Human–Animal Bonds for People Experiencing Crisis Situations

**DOI:** 10.3390/ani13050941

**Published:** 2023-03-05

**Authors:** Karl Oosthuizen, Bianca Haase, Jioji Ravulo, Sabrina Lomax, Gemma Ma

**Affiliations:** 1Sydney School of Life and Environmental Science, Faculty of Science, The University of Sydney, Camperdown, NSW 2006, Australia; 2Sydney School of Veterinary Science, Faculty of Science, The University of Sydney, Camperdown, NSW 2006, Australia; 3Sydney School of Education and Social Work, Faculty of Arts and Social Sciences, The University of Sydney, Camperdown, NSW 2006, Australia; 4Royal Society for the Prevention of Cruelty to Animals New South Wales, 201 Rookwood Rd, Yagoona, NSW 2199, Australia

**Keywords:** human–animal bonds, companion animal, pet, crisis situations, social support, coping, mental health, multifaceted

## Abstract

**Simple Summary:**

This study explores the benefits and challenges of pet ownership for people experiencing crisis situations, from the perspectives of clients from The Royal Society for the Prevention of Cruelty to Animals New South Wales (RSPCA NSW) Community Programs. Our findings were that human–animal bonds are highly valued by people experiencing crisis situations, and can affect people’s ability to seek help or refuge, and to help people recover after a crisis. Human–animal bonds provided companionship and catalyzed interpersonal connections, which improved mental health and coping during a crisis. However, separation from a pet can cause stress and anxiety, which can dissuade pet owners from seeking help. Programs such as the RSPCA NSW Community Programs address pet safety and remove cost barriers, providing relief and encouraging help-seeking. Human–animal bonds provided structure and companionship, which by improving mental health, aided in recovery, post-crisis. Additionally, the absence of a pet post-crisis negatively affects people’s recovery.

**Abstract:**

Human–animal bonds, by providing social support, have been shown to improve the health and wellbeing of pet owners, especially those experiencing a crisis situation. The human–animal bond for people in crisis situations is complex and multifaceted, as it has shown to improve health, whilst it can also discourage people from seeking help, due to fears of leaving their pet behind. The purpose of the study is to capture and to assess the role of the human–animal bond for people in crisis situations. Semi-structured interviews were conducted with pet owners involved in the RSPCA NSW Community Programs (n = 13) in 2021 and 2022. The findings of the study indicate that the human–animal bond is highly valued by people experiencing crisis situations, that the human–animal bonds can affect people’s ability to seek help or refuge, and that the human–animal bond helps people to recover after a crisis. The findings suggest that community crisis support services, prison systems, hospital systems, emergency housing, and government legislation should recognize and aim to preserve this bond to provide the best help for people experiencing crisis situations.

## 1. Introduction

Pet owners often regard their pets as friends, confidants, and family, with reference to their emotional and social intelligence [1,2]. Human–animal bonds have been associated with improvements to human health [3,4,5]. The social support hypothesis proposes that these improvements to health are the consequence of the support provided by the social connection to their pets [1,6]. Pets directly provide their owners with social support themselves, or by promoting human social interactions [2,7]. The mechanisms by which pets provide social support are hypothesized to be the catalyzation of interpersonal connections [8,9,10,11], companionship [7,8,12], and the encouragement of exercise [13]. Research has shown that the most common household pet species include cats, dogs, reptiles, fish, birds, and rodents [14,15]. These pets all provide companionship, and can catalyze interpersonal connections based on a common interest in that animal [16,17,18]. However, dogs were shown to be more likely than other species to encourage physical activity in the form of exercise and interpersonal connections through chance interactions that occur on dog walks [8].

Human–animal bonds have been associated with improvements to physical and mental health [3,4]. The biopsychosocial model conceptualizes health as the disruption or enhancement to the interrelated social, biological, and psychological factors of health [3,4]. Human–animal bonds provide social support to human beings, reducing anxiety, depression, and stress [3,19]. This in turn prevents associated physical health responses such as the hyperactivity of the sympathoadrenal medulla system and the hypothalamic–pituitary–adrenal axis, which if prolonged, can encourage chronic disease and mortality [3,19]. A broad range of investigations have found that human–animal bonds improve mental health by providing social support, which consequently improves physical health [19,20,21,22]. However, as it is possible for the mechanisms for social support to differ between pet species, their contribution to health can also vary. While cats have been shown to provide better companionship to their HIV-positive owners than dogs [23], dogs have been associated with a significant increase in exercise and enhanced physical health for their owners [24,25]. There are limited studies exploring how the species of a companion animal mediates the effect on human health [7,26,27,28].

The human–animal bond can have both positive and negative influences on the health of people experiencing crisis situations. Crisis situations are short-term overwhelming situations that disrupt an individual’s normal state, causing stress and social isolation [29]. Key crisis situations relevant to the concept of the human–animal bond are homelessness [30], domestic violence survivorship [31], incarceration [32], and hospitalization [33]. Research has indicated that the social support provided by pets is vital for the health of highly stressed or socially isolated people [8,34,35]. Recent studies have illustrated that people experiencing any of these key crisis situations tend to be socially isolated and highly stressed, and that the social support provided by human–animal bonds can be pivotal for their health and coping [36,37,38]. However, for people in crisis situations, caring for an animal can become a barrier to seeking help, as this may entail leaving their pet in a dangerous environment or surrendering them [31,39,40,41]. If a pet is left behind when escaping a crisis situation, negative cognitions associated with their pet’s safety can hinder a persons’ recovery [42].

People experiencing homelessness have profound emotional connections to their pets, and this social support is vital to their endurance in inhospitable environments [41,43]. Studies have shown that people experiencing homelessness would rather forgo accommodation than relinquish their pet to a shelter indefinitely [41,43]. Similarly, animal intervention programs have illustrated that a human–animal bond can motivate inmates to reduce recidivism and inmate violence, while promoting the development of life skills, a sense of self, and healthy behaviors, which improve coping and recovery after leaving jail [32,33,44,45]. While there is limited literature assessing the impact of the human–animal bond on hospitalization, it has been shown that human–animal bonds are associated with improved cardiovascular recovery post-surgery [46,47]. Furthermore, the responsibilities of pet ownership can delay an individual seeking healthcare [46,47]. Domestic violence survivors’ pets are crucial supporters, confidants, and allies [31,39,48,49]. However, domestic violence perpetrators can use violence against pets as a coercive control measure, making the pet a barrier to escaping a domestic violence situation [31,39,48,49].

The Royal Society for The Prevention of Cruelty to Animals New South Wales (RSPCA NSW) provides a range of community programs that provide temporary housing and care for pets whose owners are escaping crisis situations, allowing the pet owners to focus on their own wellbeing, recovery, and safety [50]. Within the program, pets are placed in foster homes or emergency boarding through private pet boarding facilities or RSPCA NSW shelters for up to three months while their owner seeks refuge or recovers [50]. These community programs also provide personalized case management, financial assistance, and access to veterinary treatment [50].

Our study aims to develop a better understanding of the role of human–animal bonds for clients of the RSPCA NSW Community Programs experiencing crisis situations, through a qualitative research approach [51,52]. This enhanced understanding will be used to better inform policy and service provision. This qualitative research creates the foundations of future investigations, and was developed to understand its research subject, rather than investigate predicted outcomes. Inductive studies have revealed nuanced insights into the accounts of people experiencing homelessness, domestic violence, and incarceration [30,53,54,55]. To the author’s knowledge, there are currently no studies that holistically explore and characterize the role of the human–animal bond for people in crisis situations by capturing both the positive benefits to health, and the negative effects on help-seeking behaviors. Our study aims to fill a gap in the existing literature by exploring the perspectives of people in crisis, to create a comprehensive and multifaceted understanding of the role of human–animal bonds for people experiencing crisis situations.

## 2. Materials and Methods

### 2.1. Sample

All clients from the RSPCA NSW Community Programs were eligible to participate in the study, and they were invited by their case worker to participate as they were discharged from the program. Clients from three Community Programs were eligible to participate: (1) Domestic Violence, (2) Aged Care, and (3) Emergency Boarding and Homelessness. Participation was optional and was with informed consent.

### 2.2. Semi-Structured Interviews

The semi-structured interviews were conducted to capture the interviewees’ perspectives on the role of the human–animal bond for people experiencing crisis situations (Table 1). Interviews were conducted in two rounds and were completed via telephone by two members of the research team (GM and KO). The first-round interviews were conducted from March to May 2021. The second-round interviews were conducted between August and September 2022.

### 2.3. Qualitative Data Analysis

Interview transcripts were subjected to a thematic analysis using an inductive approach. All interviews were transcribed in full, using the transcription tool provided by Microsoft Word. The interview audio was subsequently reviewed manually to correct any mistakes to the transcription. To analyze the interview transcripts, the data were coded using the computer-assisted qualitative data analysis software NVivo (released in March 2020) [56]. The coding process was treated as a non-linear system and re-reviewed frequently to ensure that the interpretation privileged the perspectives of the respondents. The transcripts were manually assessed, line by line [57]. Using NVivo, codes (subthemes) were created by attaching annotations, concepts, and thematic domains to specific words and excerpts from the interview transcript [57]. Subsequently, relationships were identified between the created codes (subthemes), and these relationships were used to aggregate closely interrelated codes (subthemes) to develop major codes (themes). Afterwards, major codes (themes) were selected and integrated into meaningful summaries that address key components of the role of the human–animal bond for people in crisis situations. This complete non-linear coding process is depicted in Figure 1. Themes were then reviewed to determine whether they were distinct to the clients from a particular program (e.g., Homelessness) or a specific pet species, and addressed accordingly. Major codes and summaries were then organized to build a comprehensive report of theories that address the role of the human–animal bond for people experiencing crisis situations, as depicted in Figure 2.

### 2.4. Ethical Approval

Ethical approval was granted by the University of Sydney Human Research Ethics Committee (Project Number 2020/856).

## 3. Results

Thirteen clients participated in the study; one client who experienced domestic violence, one client who experienced homelessness and domestic violence, nine clients who had been hospitalized (six from the Aged Care program and three from the Emergency Boarding program), and two clients who had been incarcerated. Of these same 13 clients: seven clients had dogs, and six clients had cats.

The thematic analysis identified three major themes; (1) the human–animal bond is highly valued by people experiencing crisis situations, (2) human–animal bonds can affect people’s ability to seek help or refuge, and (3) the human–animal bond helps people to recover after crisis. Each theme is made up of 2–5 subthemes, which explore significant elements of their main theme. The themes and subthemes identified from the interviews related to the role of the human–animal bond are reported below, with illustrative quotes. Pseudonyms have been used to protect the identities of interview participants and their animals.

### 3.1. The Human–Animal Bond Is Highly Valued by People Experiencing Crisis Situations (Theme 1)

Clients expressed that the human–animal bond was highly valuable during their crisis experience. Participants clearly identified that at the time of crisis, their animal’s companionship improved and bolstered their mental health and ability to cope. Human–animal bonds also catalyzed interpersonal connections during crisis situations, which improved mental health and the ability to cope. In contrast, the absence of a pet during the crisis negatively affected mental health and coping.

#### 3.1.1. Pet Companionship during a Crisis Situation Improves Mental Health and Coping (Theme 1, Subtheme 1)

Participants reported that their animals provided social support in the form of companionship during their crisis situation. Participants’ animal companions provided non-judgmental and unconditional love, which alleviated the social isolation caused by their crisis situation. Fraternal, friendly, and familial descriptors illustrate the strength and closeness of these connections.


*“Gosh, she’s basically the only friend I have in the world. I don’t have anybody else, I have Rosie—it’s wonderful. Rosie listens to my sad stories, she listens to my happy stories. She jumps around joyously and barks when I’m happy and sits there and mopes with me when I’m not”.—Barney*



*“Well, that’s what all animals do, unconditional love”.—Susan*



*“Ah, everything. It’s just like a child. Just like one of my kids. I mean, of course my children are people, they’re humans, but he thinks he’s a human”.—Anne*


Participants reported how companionship was associated with improvements to their mental health. Companionship from their pet was described as mentally grounding, alleviating stress and anxiety, and renewing mental strength, which allowed individuals to persevere and to cope with their crisis situation and life.


*“She’s registered with the local council as a companion, and she’s been a constant companion and she’s been a constant form of support and she gives me so much occupational therapy that I actually rely on her and she’s only a cat”.—Sara*



*“When you’re in the situation I was in, you’re sort of in a crisis situation—you’re not connected to anything, you know? You’re quite alone and even though there are services around you and that sort of stuff. The cats kept me focused on what I needed to do”.—Susie*



*“Well, it helps alleviate, that stuff <stress, depression, and anxiety>, like a great deal of it, yes”.—Sole.*


#### 3.1.2. Interpersonal Connections Catalyzed by Pets Improve Mental Health during Crisis Situations (Theme 1, Subtheme 2)

A participant who was incarcerated reported how pets can create social connections during a crisis situation. She described how these interpersonal connections were associated with improvements to mental health, by relieving stress and by being associated with positive cognitions such as joy and happiness.


*“I must have sounded freaking insane because you know I would make stories up about them, like them being an Olympic champion and stuff like that and my girlfriends would laugh. And I’d say, “I wonder what they’re doing now?”.—Susie*


#### 3.1.3. Separation from Pets during Crisis Can Cause Stress and Anxiety (Theme 1, Subtheme 3)

Participants described how the absence of their pet while they were in a crisis situation caused them to be stressed or anxious because they were concerned about their animal’s safety or wellbeing, and that this was detrimental to their mental health and physical health.


*“I was actually very, very distraught without him because he’s the one I talk to and tell things to because he doesn’t tell anybody else and he’s not judgmental. He’s just awesome”.—Anne*



*“When I’m not with her when I’m in hospital, I’m always worrying about how when she’s like kept in the kennel with the RSPCA, I’m always worrying about her and asking about her. Asking about her and sometimes call the RSPCA and see how she is. Yeah so, I will always take care of her, and they will always take good care of her”.—Sole*



*“I get very lonely without them. I just spent seven weeks in hospital, and I hate to put them into RSPCA to look after them for me and I really missed them”.—Daniel*



*“But that’s why straight away I got organized with the nurses they contact <RSPCA> and they were able to pick her up. Because when I’m worried, I’m stressed it’s not good for my heart to get stressed, they said it’s not good. That’s the problem I have. But I’m handling it. I can look after her still you know?”.—Gavin*


### 3.2. Human–Animal Bonds Can Affect People’s Ability to Seek Help or Refuge (Theme 2)

Human–animal bonds can affect people’s ability to seek help or refuge. Clients expressed how the safety of their pet was a key determinant in deciding to seek help, and that without the assured safety of their pet, they would not carry out help-seeking behaviors; for example, organizing to have surgery or seeking accommodation. A domestic violence survivor expressed how their pets were used as a form of coercion and control to prevent help-seeking behaviors, or the animals were used by the abuser to exact revenge. The RSPCA NSW and its community programs provided a trustworthy service which assured the safety of their pets and arranged payment programs which provided relief. This allowed participants to focus on seeking help or refuge for themselves.

#### 3.2.1. Pet Ownership Can Dissuade Help-Seeking Behaviors (Theme 2, Subtheme 1)

Participants described how their fear of losing or abandoning their pet made them reluctant or unable to seek help. Clients identified that they had a responsibility to their pets in looking after them, and therefore, seeking help that required surrendering them was associated with guilt and heartache. Clients reported that they would rather forgo necessary treatments or stay in unsafe situations, rather than lose their animal.


*“I had to have spinal surgery and of course I couldn’t afford the $55 per night to have [her boarded]. So, it was either find someone who can look after her for next to nothing or surrender her and the second option just did not appeal to me in any way, shape, or form”.—Barney*



*“We all love our pets and the last thing that most of us want is to have to part with them because we’re put in a situation for a while. That’s the worst, because then we’ll find a house and it’ll be like “our dog is gone””.—Anne*



*“I know it’s forever. It’s not, I’m not going to like you next week so I can give you back. The commitment you make to a dog has to be forever or not at all. That’s my view anyway”.—Angela*



*“Yeah, it’s just. It used to. I don’t know, just used to get me a bit quite depressed over the number of times I’ve had to be going now to hospital and. Yeah, it’s just the upset it creates in my house every time I go ‘cause, I’ve always gotta find somebody to look after my dogs or find somewhere that’ll take them while I’m in the hospital. And then I’ll worry about them all the time I’m in hospital”.—David*


Participants who were domestic violence survivors specifically identified how domestic violence perpetrators can use pets as a form of coercion and control to prevent help-seeking behaviors or exact revenge.


*“No [my perpetrator had not harmed my dogs previously], not that I’m aware of. No, definitely not. But he would lash out at me, basically. But I wasn’t at home, so the next best thing was something that he knew that I really loved—my animals. He knew that’s how he could get to me”.—Daisy*


#### 3.2.2. RSPCA (Theme 2, Subtheme 2)

Participants identified that RSPCA NSW was a key factor in encouraging the decision to seek help by keeping them connected to their pets and providing a payment plan. Similarly, participants reported that the assured safety of their pet provided by RSPCA NSW was associated with relief and the ability to focus on escaping from crisis and their own recovery. The safety of their pet was a major priority and requirement in the help-seeking process.


*“Really really happy. You have no idea how happy I am. And everyone I know, all my girlfriends all said I can’t believe that you actually kept the cats. I said, you know if I’m paying this off until I’m 80 it doesn’t matter. Right? Because what needed to happen, happened”.—Susie*



*“Yeah, yeah. I mean, ‘cause, we spent so much time together. No, it was great when <my RSPCA case worker> sent me the photos over. I was like, I know she’s OK. I know she’s OK. You’re looking after her, OK, looking out, or someone is looking after and she’s not here by yourself all the time. That’s what I was most worried about that she was here by herself”.—Clementine*


Participants identified how the cost of boarding was associated with an inability to seek help, and how RSPCA NSW providing a payment program helped them to overcome this challenge.


*“Thank God, RSPCA is around, I can’t believe the program, yeah it just covers everything it really does. It’s good… I just walked out of the RSPCA that day and said “yeah cool, OK”…. I knew that they’d be safe 24/7”.—Susie*



*“Because there would be an awful lot of people like me who adore animals who couldn’t afford to pay $50 a night who needed surgery who would have had to either not have the surgery or release the animal to its fate, which usually little animals like that don’t have a very good fate. So, there you go. But we don’t have to go there because the RSPCA solved that problem for me”.—Barney*


### 3.3. Human–Animal Bonds Help People to Recover after Crisis (Theme 3)

Clients expressed throughout their interviews that their relationships with their animals were important to their recovery after their crisis experience. Clients expressed that their relationship with their animals provided routine and companionship, and catalyzed connections with other people. Social support in general bolstered and improved their mental health, which aided recovery.

#### 3.3.1. Pet Ownership Encourages Structure and Routine, Which Improves Recovery and Mental Health after Crisis (Theme 3, Subtheme 1)

Participants reported that their relationships with their animals were important for their recovery from the crisis situation, because pet-related routines such as walking, feeding, or playing; and even basic chores such as cleaning food bowls and litter trays, provided structure to their life, post-crisis. Respondents identified how the requirements and routines associated with pet ownership motivated them to get through the day, and provided a sense of purpose.


*“They made me come home every night. Be there to feed them at you know—oh Lucy gets a bit whiny if it gets to about 10 past 4—she’s like a Labrador! But you know, be fed by 6 o’clock every single dad. I’m up every single morning to put food in their bowls. Their bowls are washed out religiously. These are the things I’ve got to do every day. That’s what the cats mean to me. They basically took the place of an antidepressant”*
*.—Susie*



*“Well, sometimes when I don’t feel like going out and I can’t be bothered going out today. I look at them and I think Oh, well I’ve gotta go. I’ve gotta go, gotta. I’ve gotta go and I’ve got to do this and that for them”.—Clementine*


Participants reported that the structure and routines were therapeutic, and encouraged self-care behaviors. These self-care behaviors were described to have improved mental health by reducing stressors and improving mood, which aided recovery, post-crisis.


*“Well, they keep me on my toes and always give me something to do. And when you think of occupational therapy, that’s just like second nature when you’ve got a pet you care for. It is an occupation in a way. It’s not a chore but it’s something you’ve got to do. It gives you an outlet…. Yeah! Structure! That’s the word, yeah yeah. I know it’s very beneficial as a therapy”.—Sara*



*“You know if you’re stressed out, you might stay home. I might just want a lie around but I’m not that kind of guy. I like to get up early in the morning. Sometimes 5–6 o’clock. Get Bella breakfast for her. Yeah, it makes my day, you know? For me and her. You know some people stay in bed, they are done with that and don’t care. But I care about Bella. I care about her, so I care about myself!”.—Gavin*


#### 3.3.2. Pet Companionship Improves Recovery and Mental Health, Post-Crisis (Theme 3, Subtheme 2)

Participants reported that the companionship provided by their animals was important to their recovery from their crisis situation. The presence of the animal was important to their wellbeing and recovery. Post-crisis participants identified that animal companionship was connected to their sense of self, and was like sharing their life with a person.


*“They’ve been my companions for the last 6–7 years, it’s the reason why I get up in the morning to be honest… It wasn’t all lost, like the girls were coming with me and that made it sort of better… [My cats] are the reason I feel carpet under my feet in the morning. Over the years my life got to the point where I’ve lost so much, and not just material stuff and money, but all the other shit that goes with it, and [my cats] were my anchor”.—Susie*



*“I just hope that I can keep going maybe say another 10 or 12 years, which will probably be roughly the life expectancy of a dog like Rosie and if that works out we’ll both hopefully go somewhere at the same time. Hopefully together”.—Barney*


Participants reported how this companionship was associated with mental health improvements by attenuating stressors, helping with the recovery of mental health post-crisis, and renewing internal strength and the desire to keep going with life.


*“Oh of course it does [improve my mental health]! My goodness gracious without a dog I really wouldn’t be here. I actually wouldn’t want to be here”.—Barney*



*“Well, I just appreciate them even more than I already did. It made the bond stronger between us, our little relationship, and I’m glad that they have each other as well. Because they helped each other through the healing process”.—Daisy*



*“They give me a pet to cuddle. They give me something to care for, to look after. I think that as you grow older—because I looked after my elderly parents for 12 years—as you get older you need something to look after, it keeps you going. They looked to me for food. I feed them, I cuddle them. I see to their hygiene. And they give me some sort of purpose in life. And that is so important as you get older. Something to do, someone to look after, someone to tend to. So you’re not sitting, fixating on yourself all the time. Sort of like a diversional therapy”.—Sara*


#### 3.3.3. Interpersonal Connections Catalyzed by Pet Ownership Improve Mental Health and Recovery Post-Crisis (Theme 3, Subtheme 3)

Participants reported that interpersonal connections were promoted by human–animal bonds, post-crisis.


*“If you do have an animal or pet [it] increases your chance of having longer lasting relationships with a companion in the future”.—Paul*



*“Friends on the internet on Facebook, no there’s lots of different Facebook pages. It’s great, great. Not a great amount just personal friends on Facebook, but I’ve never met. But you know, really close because we had the same condition, some same health condition”.—Sole*



*“We started chatting one day about a year ago I guess when she sort of pulled me up when I was taking Rosie for a walk… But every afternoon, 2–3 km she used to take her and Rosie loves that and that’s something that I can’t do. So that ‘fit’ is very very good for all of us”.—Barney*


Participants reported how interpersonal connections were associated with mental health improvements, because human connections created because of their relationship with their animal promoted positive emotions and elevated mood.


*“All the community, the people inside the community here, some of them get along with her, mostly all of them. The ones who are trusted and a few friends”.—Gavin*



*“People stop with their dogs… kids are coming up… Everybody knows him. Probably ‘cause I’ve got a big mouth”.—Simone*



*“<It keeps me> mentally healthy, you know we laugh, we share funny names and pictures of cats and things like that. Cats doing strange things yeah”.—Sole.*


#### 3.3.4. Pets Encourage Physical Exercise, Post-Crisis (Theme 3, Subtheme 4)

Participants who owned dogs regularly reported how their pets encouraged them to go on walks in a post-crisis setting.


*“I push her to the park, but whenever she [was] in the park I let her go off-leash you know, let her run around and we go and do some shopping and come back with the stroller with her. It takes me like 1 h to the park, 1–2–3 h. It makes my day like when I have nothing to do. It makes me feel relaxed. You know stressed out all gone. Do you understand?”.—Gavin*


A participant identified how their cat motivated them to care for their own physical health by discouraging smoking.


*“That’s why I gave up smoking because they won’t come near you, and why would they?”.—Susie*


Despite a thorough discussion of physical health, clients frequently discussed physical health in terms of how much exercise their pet was encouraging them to do, rather than the physical condition of their body.


*“Physical health… I don’t really go on as much walks because I’ve been struggling to breathe because of the chest infection”.—Paul*



*“I don’t move around well so it’s not much physical exercise, but mentally yeah, they’ve been absolutely great”.—Arthur*


#### 3.3.5. Absence of Their Pet, Post-Crisis, Attenuates Recovery and Mental Health (Theme 3, Subtheme 5)

Participants identified how the absence of their pets would be associated with worse mental health and limited recovery, commonly connected to a lack of desire to live.


*“If I hadn’t had the dogs here, I don’t know what would have happened. I think I just would have cut the top, the bundling, lost the plot completely. I just wouldn’t have been able to cope”.—Daniel*



*“I probably wouldn’t’… I probably would be dead if I didn’t have her. She’s my lifeline”.—John*


## 4. Discussion

This explorative study examined the role of the human–animal bond for people experiencing crisis situations. Our thematic analysis of semi-structured interview transcripts identified three major themes: (1) human–animal bonds are highly valued by people experiencing crisis situations, (2) they can affect people’s ability to seek help or refuge, and (3) they can help people recover after crisis. These major themes identify three distinct phases where human–animal bonds are important for people experiencing crisis.

### 4.1. Human–Animal Bonds Are Highly Valued by People Experiencing Crisis Situations (Theme 1)

A key aspect of the role of human–animal bonds for people experiencing crisis situations is its ability to provide companionship and to catalyze interpersonal connections, which by improving mental health enhances an individual’s ability to cope with the crisis. This highlights the importance of keeping pets connected to their owners throughout and beyond their crisis situations.

Companionship provided by animal companions was found to be important for those experiencing crises, by providing non-judgmental and unconditional love, that alleviated the social isolation caused by their crisis situation. This finding corresponds with existing literature that has illustrated that people experiencing domestic violence [58,59] and homelessness [41,60] felt isolated and emotionally distressed. Furthermore, these studies highlighted how the non-judgmental and unconditional love provided by pet companionship was a profound form of emotional support that could attenuate these feelings of isolation and emotional distress [41,58,59,60].

Our results indicate that animal companionship improved mental health, allowing individuals to persevere and to cope with their crisis situation and life. This result supports a broad range of investigations that illustrate that companionship provided by human–animal bonds can improve mental health by reducing anxiety, depression, and loneliness [7,8,19,41]. Cleary et al.’s (2020) integrative review found that people experiencing homelessness described how animal companionship improved their mental health and their ability to cope with homelessness. Overall, our findings extend contemporary understandings of animal companionship by illustrating how companionship attenuates social isolation, improves mental health, and is an invaluable part of coping for people experiencing all crisis situations.

Our study found that for a participant who was incarcerated, pets catalyzed social connections during their crisis situations. These interpersonal connections were associated with improvements to mental health, by relieving stress and by causing feelings of joy and happiness. To the author’s knowledge, there is limited research investigating the importance of interpersonal connections for people experiencing crisis situations. However, this finding supports the literature that has demonstrated that pet ownership can act as a common quality that facilitates the creation of social support networks which protect against social isolation and that support the release of negative emotions which improve health [10]. Our result of only being described by an individual experiencing incarceration is most likely due to the fact that during crisis situations such as hospitalization, homelessness, and domestic violence, making connections with new people is very difficult. This finding should be explored appropriately in future studies.

Our findings demonstrate that the absence of a pet while a person is in a crisis situation caused them to be stressed or anxious, and that this was detrimental to their mental and physical health. A wide range of literature has similarly described that the fear of losing pets is highly traumatic and is associated with anxiety [10]. Our findings extend this understanding to illustrate how the absence of a pet during a crisis intensifies fear and anxiety, and can make people focus on the uncertainty of the wellbeing of their pets.

### 4.2. Human–Animal Bonds Can Affect People’s Ability to Seek Help or Refuge (Theme 2)

Pet owners experiencing a crisis situation are reluctant or unable to seek help or refuge because they fear losing or abandoning their animal. Hence, it is important to include pets and to consider them as part of the family unit when helping people to access services during a crisis situation. The RSPCA NSW’s Community Programs were a key factor in encouraging the decision to seek help by keeping pet owners connected to their pets and by providing a payment plan, allowing them to focus on themselves. This result supports a range of studies that have illustrated that the fear of losing a pet can dissuade people experiencing homelessness [30], domestic violence [49], and hospitalization [61] from seeking help.

Our results identified that domestic violence perpetrators can use pets as a form of coercion and control to prevent an individual seeking help or to exact revenge. This finding is corroborated by Haden et al. (2018) and Hardesty et al. (2013) that found that the abuse of companion animals is used as a coercive tactic by domestic violence perpetrators to control their partners by creating an environment of fear, dependency, and helplessness [49,62].

The current study found that the RSPCA NSW provided a trustworthy service that by assuring the safety of their pet and arranging payment programs, provided relief, allowing people to focus on help-seeking behaviors. This finding extends previous research by highlighting how help-seeking behaviors can be encouraged by keeping people connected to their pet.

### 4.3. Human–Animal Bonds Help People Recover after a Crisis (Theme 3)

A key benefit of human–animal bonds during crisis recovery are their ability to improve mental health and wellbeing by providing structure and routines, and companionship, and catalyzing interpersonal connections. The importance of pets encouraging routines and structures has been identified in research investigating the human–animal bond and elderly people [8]. Hui Gan et al. (2020) highlighted how pet-related routines such as feeding, playing, and grooming provide structure to the lives of elderly pet owners. We found that this benefit was consistent amongst the clients of all the RSPCA NSW programs, and not just for elderly clients.

Our results indicate that the structures and routines provided by human–animal bonds were therapeutic and encouraged self-care behaviors that improved mental health and recovery. This finding is supported by Hayden-Evans et al. (2018), which found that pets, by encouraging routine and structure, can provide a life purpose that aided in the development of recovery skills. There is limited research investigating the importance of routine for people experiencing crisis. However a key challenge for people post-crisis is recovering in an unfamiliar and confusing environment; therefore, pet-related routines are valuable in providing structure to navigate this new environment and to enhance recovery [29].

Our study found that human–animal bonds were important for a recovery from crisis situations by providing companionship, which was connected to a sense of self. To our knowledge, there are no studies that identify the importance of animal companionship for an individual’s sense of self, post-crisis. However, the importance of animal companionship to a sense of self has been described in studies of people in aged care. Hui Gan et al. (2020) highlighted how the importance of the human–animal bonds for people in aged care is connected to their perception of themselves and how they are to spend the rest of their life with another living being. Therefore, this finding extends the definition of animal companionship for people experiencing crisis situations to be connected to a sense of self and sharing a life with their pet.

Our findings demonstrate that animal companionship by attenuating stressors was associated with mental health improvements, recovery post-crisis, and renewing internal strength and the desire to keep going with life. Studies such as Cleary et al.’s (2020) have found that people experiencing homelessness described how animal companionship was important to their recovery after finding accommodation because of positive effects on their mental health. However, there are currently no studies that identify the importance of animal companionship to a sense of self and the improvement of mental health. Overall, our findings extend contemporary understandings of animal companionship by illustrating how companionship is connected to a sense of self, improves mental health, and is an invaluable part of recovery for people experiencing all crisis situations.

Our results indicate that human–animal bonds promote interpersonal connections post-crisis, and that this contributed to improved mental health. This finding supports the literature that demonstrates that pet ownership can act as a common quality that facilitates the creation of social support networks [10]. Social support networks protect against social isolation and support the release of negative emotions, which improve health [10]. Studies have shown that having a larger number of interpersonal connections or friends was associated with better mental health, and that this is connected to the creation of social support networks [63]. Socially isolated individuals may still struggle with making connections in person, and participants highlighted the importance of online communities such as Facebook. This finding is corroborated by Newman et al. (2019) who found that online communities such as Facebook or online forums can allow socially isolated individuals to regain a sense of social inclusion and belonging. Unlike during crisis situations, for people recovering from crisis situations, this finding was described by all participants, demonstrating that in a post-crisis environment, people can more effectively create interpersonal connections promoted by their pets.

Our findings indicate that post-crisis, dogs encouraged their owners to go on walks, whereas cats did not. Research has shown that dogs are more likely to encourage physical activity in the form of exercise than other common pet species [8]. Furthermore, pets dissuaded cigarette smoking which would improve physical health. To this author’s knowledge, this paradigm has not been explored and should be explored in future studies. A noteworthy, unexpected aspect of this finding is that participants did not identify any direct influences on their physical health. Despite thorough discussion, most participants interpreted physical health to mean how much exercise their pet was encouraging them to do, rather than the physical condition of their body. This could be due to the fact that social media and pop-culture reinforce an understanding of physical health to be connected to exercise and physical activities, such as going running or going to the gym [64]. This finding was unlike a broad range of investigations that found that the human–animal bond improved physiological health markers such as blood pressure, serum triglycerides, heart rate variability, and cholesterol levels [19,20,21]. The lack of responses indicating improvements to physical health could also be because qualitative studies are inadequate at perceiving the quantitative variables in the beforementioned studies that indicated the physical health of the body. Therefore, future studies could attempt a mixed-methods model to better address physical health. Ultimately, as descriptions of physical health were scarcer and more limited than mental health, this finding alludes to the fact that people experiencing crisis situations find that the role of human–animal bonds is more focused on mental health improvements rather than on physical health improvements.

Our study found that the absence of a pet post-crisis was associated with predictions of worse mental health and connected to suicidal ideation and limited recovery. This reflects commonly reported phenomena in the crisis literature where people experiencing homelessness and domestic violence highlight that the absence of their pet would take away their life purpose and attenuate their recovery from their crisis situation [31,49].

## 5. Limitations

A key limitation of the study is the possibility of sampling bias. A person who is going to accept an interview from the RSPCA NSW is likely going to have successfully, coped, escaped, and recovered from their crisis situation because of the RSPCA NSW Community Programs. Therefore, participants in this study were likely to endorse the RSPCA. Future studies could improve understandings of the human–animal bond for people experiencing crisis situations by interviewing people who did not successfully cope, seek help, or recover from crisis situations, and contrast this with people who were successful. There are unequal numbers of clients from different crisis situations, which may have led to the overrepresentation or underrepresentation of certain phenomena. Future studies should attempt to consolidate the perspectives of people who have experienced incarceration, homelessness, and domestic violence. Participants in this study tended to be older than 60 years of age and described phenomena common to research investigating human–animal bonds and elderly people. Future studies should investigate participants from a variety of age groups to determine whether the described phenomena are due to the ages of the participants. Participants in this study only owned cats and dogs, limiting the studies’ generalizability to the roles of all pets. Companionship and the catalysis of interpersonal connections are not unique to only dogs and cats [16,17,18]. Future studies could explore more pet species to see if there are differences in the way in which particular animal species provide social support. Our study found that participants interpreted physical health to mean how much exercise their pet was encouraging them to do, rather than the physical condition of their body. As this was a qualitative interview-based study, physical measures used in conceptually similar quantitative studies such as platelet activity, triglyceride levels, and hyperactivity of the hypothalamic–pituitary–adrenal axis could not be measured. Therefore, future studies could attempt to incorporate a mixed-methods model that incorporates both the perspectives of individuals in crisis situations and also the physical quantitative measures of their health.

## 6. Implications

The human–animal bond is integral for coping, escaping, and recovering from crisis, and the social support that it provides is crucial for the mental wellbeing of their owners. This finding has important real-world applications. In Australia, most domestic violence shelters, homeless shelters, and rental accommodation do not accept pets [65,66], resulting in animal shelters being inundated with pets surrendered by owners who are unable to secure affordable or pet friendly accommodation [65]. Furthermore, in Australia, a large percentage of individuals who had experienced crisis situations did not know about services that would provide them with aid [65]. The findings of this study suggest that crisis support services, prison systems, hospital systems, animal welfare/health organizations, and government departments and agencies should collaborate to keep owners connected to their pets throughout their crisis situation, or incorporate better systems for managing pets. Moreover, government policy and community-based programs should aim to consider pets as part of the family unit in their planning or management of people in crisis, as the described impacts to their health and willingness to seek help could be fundamental to their health and survival. Programs such as the RSPCA NSW Community Programs protect the vital connections between pet owners and pets, and ultimately serve as a lifeline for the coping, escape, and recovery of people experiencing crisis situations.

## 7. Conclusions

This study, by investigating and analyzing the accounts of people experiencing crisis situations, has created a comprehensive framework for understanding the role of human–animal bonds for people experiencing crisis situations. The social support of human–animal bonds provided through companionship and interpersonal connections is fundamental to the mental health and coping abilities of people during crisis situations. A fear of losing their pet or endangering them can discourage help-seeking behaviors. However, the provision of appropriate resources, in our case, through the RSPCA NSW, that mediate costs and that keep the individual connected to their pet, can encourage help-seeking behaviors. The social support of human–animal bonds provided through structure and routine, and through companionship and interpersonal connections, was important for mental health and their recovery from crisis. In contrast, the effects on physical health were inconclusive and should be explored further. Ultimately, the findings from our study clearly highlight the importance of keeping pet owners connected to their pets throughout and beyond their crisis experience. Services such as the RSPCA NSW Community Programs should be supported and expanded. Crisis support services, prison systems, hospital systems, animal welfare/health organizations, and government legislation should attempt to preserve and to strengthen the human–animal bond to best support the wellbeing, coping, and recovery of people experiencing crisis situations.

## Figures and Tables

**Figure 1 animals-13-00941-f001:**
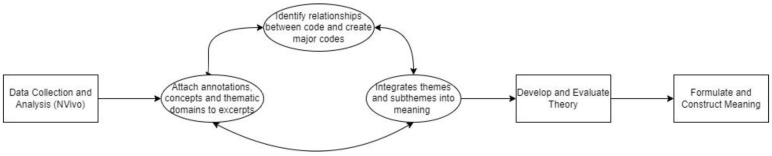
Non-linear qualitative coding process diagram.

**Figure 2 animals-13-00941-f002:**
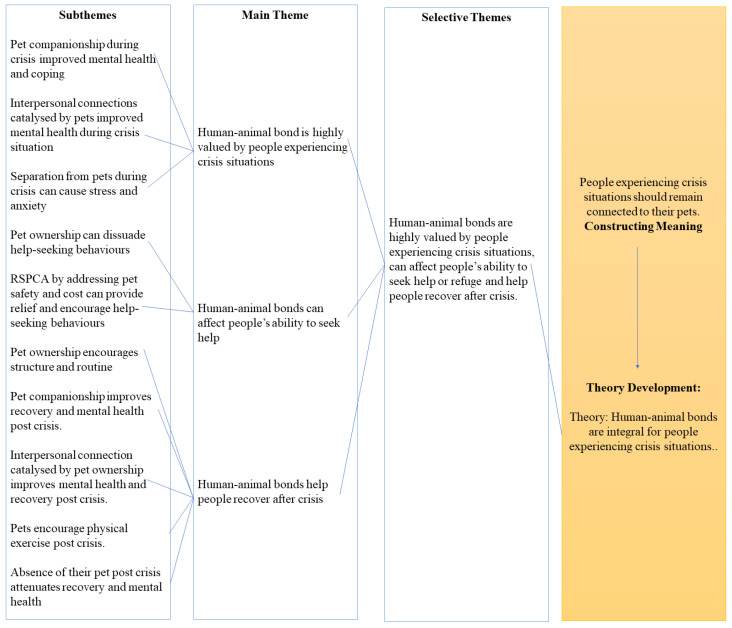
Qualitative coding and theory development diagram.

**Table 1 animals-13-00941-t001:** Examples of semi-structured interview questions.

**Questions**
What companion animals do you care for, and what do they mean to you? Can you tell me a bit about life before [ANIMAL] came to RSPCA? What is life like now? What difference has RSPCA made for you? What is the biggest difference in your life now? How has having a pet affected your health? What value could you put on this change? (Asked for a stated preference) What aspect of pet ownership is most important to your day-to-day life? What aspects and influences in your life impacted your decision to seek help? How much of this difference in your life is due to RSPCA? Can you estimate a percentage? What would have happened if it weren’t for RSPCA? Are you aware of other organizations offering a similar service to the RSPCA program?
**Secondary Interview Questions**
**Human–animal bonds are valuable to people during a crisis** Could you tell me a bit about your story and how you came to be involved with the RSPCA program? Could you describe some instances where your pet has supported you during your crisis situation? How would you describe your wellbeing or state of mind during this situation, did your pet influence this in any way? Can you recall any instances where your pet improved your physical health, where physical health refers to the condition of your body? **Help-seeking behaviors** What helped you seek help? How did being a pet owner influence your ability and desire to seek help? What aspects and influences in your life impacted your decision to seek help from the RSPCA? **Human–animal bonds are important to recovery, post-crisis** How has having a pet affected your health? What aspects of pet ownership are most important to your day-to-day life? How has your pet affected you after your crisis situation? How would you describe your wellbeing or state of mind after your crisis situation, did your pet influence any of that?

## Data Availability

The data presented in this study (de-identified interview transcripts) are available on request from the corresponding author. The data are not publicly available to protect the privacy of research participants.

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
