# Peer review of "The Role of Human–Animal Bonds for People Experiencing Crisis Situations"

_animals, 2023, doi:10.3390/ani13050941_

Round 1

Reviewer 1 Report

This is very well-written paper. It is ready for publication. I hope it will move forward quickly.

Line 115 - Isn't "holistic" a more widely accepted spelling?

Table 1 Question #3. Is this (comma) a typo?

Author Response

Thank you for your review of our article and for the opportunity to improve our manuscript. Please see specific responses below.

Response to Reviewer 1 Comments:

Point 1: Line 115 - Isn't "holistic" a more widely accepted spelling?

Response 1: Yes, thank you, changed line 115 from “wholistically” to “holistically.

Point 2: Table 1 Question #3. Is this (comma) a typo?

Response 2: Yes, it was, changed Table 1 Question #3 from “What is life, like now? to “What is life like now?”.

Reviewer 2 Report

Comments on the reviewed manuscript “The role of human-animal bonds for people experiencing crisis situations” submitted to the Animals

General comments

I appreciate the opportunity to review this exciting manuscript, which is an explorative study of the human-animal bonds for people experiencing crisis situations. The findings suggest that community crisis support services, including care of pets, are essential for pets and people. In the methods section, the authors describe the application of semi-structured questionnaires to 13 people. The analysis is qualitative. The study's findings are that pets provide adequate emotional support, increase sociality, and improve mental health.

I read a generalization and findings far more significant than they could be. The method is limited to an alleged postulation of the benefits of pet companionship and the care provided by the RSPCA NSW during crises.

I noticed that the authors don't seem up to date with the topics proposed in the article. The manuscript does not deal with a topic that has not been investigated in the last decade. Strangely, the authors state in line 69, "There are currently no studies or data from owners of species, other than cats and dogs, that explore differences in the effect of companion animals on human health.” I could cite dozens of articles that contradict the authors' claim, but please do a more detailed search in the scientific literature (for example, Brooks et al., 2018; Scoresby et al., 2021). Again in lines 114, 564, and 578, the authors claim to be unaware of articles that address the benefits of companion animals on human health and “individual’s sense of self”. When reading the list of references, I understood the cause of the wrong statements by the authors: the literature consulted on this topic is old. Much has been published in the last decade.

The sample size is limited to make generalizations, as the authors are well-recognized. Furthermore, the authors mention improvements in "depression," "anxiety," and "mental health", but this has not been proven in the present investigation. The statements of the 13 respondents are ambiguous and have the bias of having been assisted by the RSPCA NSW, as recognized by the authors. Thus, the sample is biased toward the study findings.

However, it is not new about the benefits of pets on people's mental health. The scope of the study appears to be limited. 

 References 

Brooks, H. L., Rushton, K., Lovell, K., Bee, P., Walker, L., Grant, L., & Rogers, A. (2018). The power of support from companion animals for people living with mental health problems: A systematic review and narrative synthesis of the evidence. BMC Psychiatry, 18(1), 1-12.

Scoresby, K. J., Strand, E. B., Ng, Z., Brown, K. C., Stilz, C. R., Strobel, K., ... & Souza, M. (2021). Pet ownership and quality of life: A systematic review of the literature. Veterinary sciences, 8(12), 332.

Author Response

Thank you for your review and for the opportunity to improve our manuscript. Please find specific responses to points raised below.

Response to Reviewer 2 Comments:

Point 1: I noticed that the authors don't seem up to date with the topics proposed in the article. The manuscript does not deal with a topic that has not been investigated in the last decade. Strangely, the authors state in line 69, "There are currently no studies or data from owners of species, other than cats and dogs, that explore differences in the effect of companion animals on human health.” I could cite dozens of articles that contradict the authors' claim, but please do a more detailed search in the scientific literature (for example, Brooks et al., 2018; Scoresby et al., 2021)

Response 1: Thank you for the suggestion of these additional references. I have clarified the ambiguity of this sentence. However, Scoresby et al. 2021 study excluded pet owners who did not own cats or dogs. We have incorporated the references and changed line 69 from “There are currently no studies or data from owners of species, other than cats and dogs, that explore differences in the effect of companion animals on human health [26]” to “There are limited studies exploring how the species of a companion animal mediates the effect on human health”.

Point 2: Again, in lines 114, 564, and 578, the authors claim to be unaware of articles that address the benefits of companion animals on human health and “individual’s sense of self”. When reading the list of references, I understood the cause of the wrong statements by the authors: the literature consulted on this topic is old. Much has been published in the last decade.

Response 2: Thank you for the suggestions.

Line 114 highlights the importance of a holistic exploration of positive health improvements in a crisis situation context which has not been explored. Similarly, Line 564 the study explores the effect on “an individual’s sense of self post-crisis”. There are definitely studies exploring the benefits of companion animals on human health and ‘sense of self’ which is acknowledged and explored in the later sections, but there are no studies of this phenomenon in the context of a crisis situation. I have clarified the ambiguity regarding this statement in Line 568. From “However, there are currently no studies that identify the importance of animal companionship to sense of self and improving mental health”. To “However, there are currently no studies that identify the importance of animal companionship to sense of self and improving mental health, in a post crisis context”.

Point 3: The sample size is limited to make generalizations, as the authors are well-recognized.

Response 3: We acknowledge the sample size as a limitation in the discussion line 628 – 637.

Point 4: Furthermore, the authors mention improvements in "depression," "anxiety," and "mental health", but this has not been proven in the present investigation.

Response 4:

Our results indicate that animal companionship improved mental health; it does not attempt to quantitatively capture the improvements in depression, anxiety, or mental health.

The main finding of this study is that study participants highlighted that animal companions provide companionship, interpersonal connections and a sense of structure and routine which they highlighted improved their mental health, coping and recovery.

Here are a few examples where clients have mentioned improvements to their mental health.

“It makes my day like when I have nothing to do. It makes me feel relaxed. You know stressed out all gone. Do you understand?” – Gavin

“I probably wouldn’t’ ... I probably would be dead if I didn’t have her. She’s my lifeline”.

“<It keeps me> mentally healthy, you know we laugh, we share funny names and pictures of cats and things like that. Cats doing strange things yeah”-

“These are the things I’ve got to do every day. That’s what the cats mean to me. They basically took the place of an antidepressant.

The section you may be referring to is line 504, which simply highlights how respondents’ expressions of improvements to their mental health would be in support of a range of studies that indicate that animal bonds can improve mental health by reducing anxiety, depression, and loneliness.

Point 5: The statements of the 13 respondents are ambiguous and have the bias of having been assisted by the RSPCA NSW, as recognized by the authors. Thus, the sample is biased toward the study findings.

Response 5: Yes, I agree, we acknowledge this limitation as in the discussion line 628. Additionally, sentence 633 has been added “Therefore, participants in this study were likely to endorse the RSPCA”.

Point 6: However, it is not new about the benefits of pets on people's mental health. The scope of the study appears to be limited

Response 6: Yes, this aspect is not completely novel as identified in the introduction, the novelty is its exploration of the perspectives of people experiencing crisis situations and attempting to capture a holistic interpretation of the effect of companion animals on the mental and physical health of these people.

Reviewer 3 Report

Overall, I felt this was a very strong paper and appreciate the qualitative data presented throughout. The only part of the paper that I believe could be strengthened is the introduction. Perhaps just adding transition sentences would enhance the flow/readability of this section. 

It also might be including a noted limitation in the fact that participants in this study are prone to being animal-lovers given that they used RSPCA NSW program. A future direction would be to consider these research question for a more generalized crisis-seeking audience. 

Author Response

Thank you for reviewing our manuscript and for your kind words. Please find below the response to the point raised.

Response to Reviewer 3 Comments:

Point 1 It also might be including a noted limitation in the fact that participants in this study are prone to being animal-lovers given that they used RSPCA NSW program. A future direction would be to consider these research question for a more generalized crisis-seeking audience.

Response 1: Yes, I agree, sentence 633 added “Therefore, participants in this study were likely to endorse the RSPCA”.

Round 2

Reviewer 2 Report

Dear authors

I appreciate the authors' consideration in responding to my previous comments. The explanations and modifications in the text were few but essential. I understood more the qualitative aspect of the study, although I disagree with some interpretations. However, the authors assume the risk of being questioned by other colleagues regarding interpreting the questionnaire responses. These questions are regular, and the debate helps us improve our knowledge about the relationship between humans and their pets.

Sincerely,